# An Empirical Characterization of the Aerosol Ångström Exponent Interpolation Bias using SAGE III/ISS Data

Robert P. Damadeo[1], Viktoria F. Sofieva[2], Alexei Rozanov[3], and Larry W. Thomason[1]

[1]NASA Langley Research Center, Hampton, VA, USA
[2]Finnish Meteorological Institute, Helsinki, Finland
[3]University of Bremen, Bremen, Germany

**Correspondence:** R. P. Damadeo (robert.damadeo@nasa.gov)

**Abstract.** This work uses multispectral measurements of vertically resolved aerosol extinction coefficient from the Stratospheric Aerosol and Gas Experiment (SAGE) III on the International Space Station (ISS) to demonstrate how the use of the Ångström exponent for interpolation of aerosol data between two different wavelengths creates a bias. An empirical relationship is derived between the magnitude of this bias and the Ångström exponent at several different SAGE wavelengths. This relationship can thus be used as a correction factor for other studies, such as multi-instrument intercomparisons or merging, that wish to convert aerosol data from one wavelength to another using the Ångström exponent and is applicable to all stratospheric non-cloud aerosol except highly aged particles that are evaporating at altitudes above the Junge layer.

## 1 Introduction

Stratospheric aerosols, which are primarily sourced from particles that are either transported by tropospheric dynamics (e.g., through strong convection from a pyrocumulonimbus (pyroCb)) or directly injected (e.g., from a major volcanic eruption) (Kremser et al., 2016, and references therein), play a major role in chemistry involving trace gas species like ozone (Solomon et al., 1996) and are a key component of the Earth's radiation budget (Solomon et al., 2011; Schmidt and Robock, 2015). Given the impact of aerosols on chemistry, weather, and climate, the accuracy of climate models depends upon a reasonable representation of aerosol either through the incorporation of observationally based global climatologies (Thomason et al., 2018) or through interactive modeling of variations in aerosol. Aerosol climatologies require combining observations from different instruments using different observation techniques at different wavelengths into a single record (Kovilakam et al., 2020; Sofieva et al., 2023a). Converting aerosol extinction measurements from one wavelength to another requires knowledge of the spectral behavior of aerosol, which is dependent upon its composition and the particle size distribution (PSD). Knowledge of the PSD is also important for both calculating aerosol extinctions using Mie theory (Malinina et al., 2018; Wrana et al., 2021; Pohl et al., 2023; Knepp et al., 2024) as well as for retrieval of aerosol from limb scatter measurements (von Savigny et al., 2015; Rieger et al., 2018, 2019; Taha et al., 2021; Zawada et al., 2018; Rozanov et al., 2024). The most commonly assumed spectral dependency of aerosol extinction is that of the Ångström exponent (AE) (Kar et al., 2019; Kovilakam et al., 2020), though the curvature of the AE shape and that of an actual "aerosol spectrum" (i.e., extinction as a function of wavelength) do not exactly match (Thomason et al., 2010). It has been previously shown that the use of AE for wavelength conversion can create

biases (Rieger et al., 2015; Malinina et al., 2019), though these studies relied on using Mie theory, and assumptions about the aerosol, to compute extinction values to quantify the impact. Herein we use multispectral observations of the aerosol extinction coefficient to characterize biases induced by the use of the AE for wavelength conversion under a range of aerosol loading conditions and compositions as well as suggest an empirical relationship between that bias and the AE itself. This empirical relationship can then be used as a correction factor for efforts to convert stratospheric aerosol data between wavelengths, such as is done for intercomparison of aerosol from different instruments or for the creation of multi-instrument aerosol climatologies.

## 2  Data and Usage

While there are many different remote sensing instruments that have observed or currently observe stratospheric aerosol (hereafter simply referred to as aerosol for brevity), only the occultation measurement technique, which views the target (such as the Sun) both above and directly through the atmosphere and compares them (McCormick et al., 1979), intrinsically provides vertical profiles of total extinction data from which aerosol can be retrieved, after accounting for Rayleigh scattering and absorption from trace gases (Chu and McCormick, 1979; Damadeo et al., 2013), without any assumptions about composition or PSD (given enough spectral channels). Assumption-free (or mostly assumption-free depending upon the instrument) aerosol data makes solar occultation measurements, with their high signal-to-noise ratio, ideal for evaluating the spectral behavior of aerosol. For this reason, we use data from the Stratospheric Aerosol and Gas Experiment (SAGE) III onboard the International Space Station (ISS). SAGE III/ISS is a solar occultation instrument, operating since 2017, that provides vertical profiles of trace gases such as ozone, nitrogen dioxide, and water vapor as well as aerosol extinction coefficient at 9 different wavelengths in the ultraviolet, visible, and near-infrared part of the spectrum. The 9 wavelengths can vary slightly (<1 nm) from event to event, but are roughly centered at the following wavelengths: 384.1, 448.7, 520.5, 601.7, 676.1, 756.0, 869.2, 1021.5, and 1543.9 nm. These aerosol profiles are retrieved from as low as the surface/cloud-top up to 45 km. The retrieval algorithm and methodology are described in detail in the SAGE III/ISS Algorithm Theoretical Basis Document (Wofsy et al., 2002) and in a more recent work first evaluating v5.1 of the ozone data product (Wang et al., 2020). The only assumption regarding aerosol used in the retrieval is that the aerosol spectrum should be slowly varying in almost all stratospheric conditions (Thomason et al., 2010).

While the data version used for this work is v5.3, some of the conclusions regarding the quality of the SAGE III/ISS aerosol data from Wang et al. (2020) still hold, in particular the presence of a "dip" in the aerosol spectrum (i.e., a negative bias) affecting the 520, 602, and 676 nm channels that is readily apparent in the stratosphere in most individual profiles. This anomaly was noted in the v5.2 release notes (https://sage.nasa.gov/wp/wp-content/uploads/2021/07/SAGEIII_Release_Notes_v5.2.pdf) and remains unchanged in v5.3. The cause of the dip is still under investigation, but it is believed to be related to a possible deficiency in the ozone cross-section database used for the retrieval algorithm (Bogumil et al., 2003), resulting in an underestimation of aerosol extinction at these channels where ozone absorption is strongest. At 602 nm (where it is largest), this bias can be as large as 20% in individual profiles (typically at lower aerosol loading above the Junge layer), but averages out to less than 5% in monthly zonal means in the Junge layer and reduces with greater aerosol loading. To avoid this negative bias in data used for this work, we compute a correction by using the results of a fit to the aerosol spectrum (similar to what was shown in

Fig. 3 of Wang et al. (2020)). A second order polynomial in log of extinction versus log of wavelength using the 449, 756, 869, and 1021 nm channels for each profile at each altitude provides interpolated fit values of aerosol at 520, 602, and 676 nm to be used for this study. Interpolating across channels to account for the dip in the aerosol spectrum is a similar process as what has been done in previous studies that use SAGE III/ISS data (Chen et al., 2020; Thomason et al., 2021; Knepp et al., 2022; Kovilakam et al., 2023).

## 3 Methodology

A primary goal of this study is to evaluate the efficacy of the use of the Ångström exponent (AE) for wavelength conversion of aerosol data. The AE, which assumes the aerosol is predominantly scattering instead of absorbing, imparts an assumed shape to the aerosol spectrum, thus often acting as an ad-hoc (albeit imprecise) indicator of effective particle size, using the following equation (Ångström, 1929):

$$\frac{k_{\lambda_2}}{k_{\lambda_1}} = \left(\frac{\lambda_2}{\lambda_1}\right)^{-\alpha}, \tag{1}$$

where $k_\lambda$ is the extinction coefficient at a particular wavelength $\lambda$ and $\alpha$ is the AE. For this study AE is evaluated using the 520 and 1021 nm channels as this is one of the most common pairs for evaluating the AE using SAGE data (particularly the historical SAGE II data record), though naturally using a different wavelength pair would result in a different value for the AE. A simple illustration of the deficiencies in using AE is shown in Fig. 1. Select monthly zonal means of SAGE III/ISS data, corresponding to different aerosol loading conditions, are shown with exes (these data are raw and unchanged). The specific events referred to here are as follows: Canadian wildfires (2017; Bourassa et al. (2019); Kloss et al. (2019); Yu et al. (2019)), Ambae eruption (2018; Kloss et al. (2020); Malinina et al. (2021)), Raikoke eruption (2019; Kloss et al. (2021); Gorkavyi et al. (2021); Knepp et al. (2022); Boone et al. (2022)), Australian wildfires (2020; Kablick III et al. (2020); Khaykin et al. (2020); Yu et al. (2021)), and Hunga-Tonga eruption (2022; Mishra et al. (2022); Taha et al. (2022); Zhu et al. (2022); Duchamp et al. (2023)). Polynomial fits to the these data (as described earlier) are shown in squares and solid lines. The "dip" in the aerosol spectrum is noticeable at times (seen as differences between the exes and squares in the 520, 602, and 676 nm channels), although diminished in these averages, showing the need for this correction in the current version of SAGE III/ISS aerosol data. The apparent large negative bias in 384 nm data (again seen as the difference between exes and squares) is actually an artifact, since at lower altitudes the signal in this channel drops significantly from molecular scattering and the instrument can no longer measure aerosol extinction. The dashed lines show what an AE interpolation between the 520 and 1021 nm channels would look like for each of these spectra, highlighting a negative bias in all circumstances.

To characterize this bias, we use the polynomial interpolated value of aerosol at 520 nm and the originally measured value of aerosol at 1021 nm to compute both the AE and the AE-interpolated value of aerosol at 756 nm ($k_{756\_int}$) using Eqn. 1, which is then compared to the originally measured value of aerosol at 756 nm ($k_{756\_meas}$). While final results will be evaluated for multiple wavelengths, 756 nm is chosen as the primary demonstration in this work because it is a wavelength reported by several other satellite-based aerosol retrievals such as the Optical Spectrograph and InfraRed Imaging System (OSIRIS; 2001–

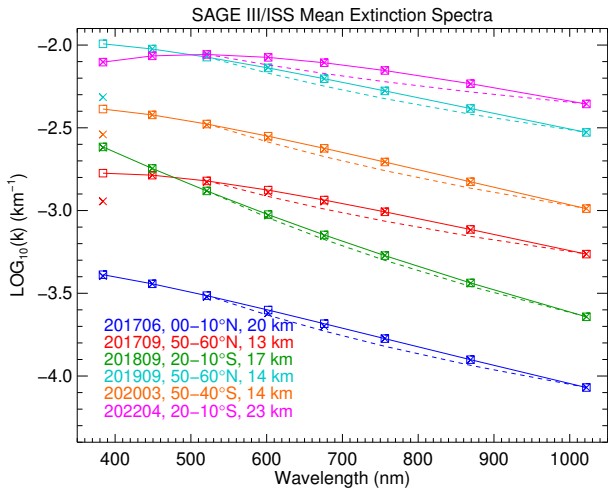

**Figure 1.** Several examples of mean extinction spectra from SAGE III/ISS. Mean data, taken over the months, latitude bands, and altitudes listed, are shown with exes. Fits to these spectra (as discussed in the text) are shown with squares and solid lines. Interpolations using the Ångström exponent are shown in dashed lines. The times and locations chosen are meant to represent, in order from top to bottom as listed in the figure text, "background", Canadian wildfires (2017), Ambae eruption (2018), Raikoke eruption (2019), Australian wildfires (2020), and Hunga-Tonga eruption (2022).

present; Rieger et al. (2019)), Global Ozone Monitoring by Occultation of Stars (GOMOS; 2002–2012; Sofieva et al. (2023b)), Scanning Imaging Spectrometer for Atmospheric Chartography (SCIAMACHY; 2002–2012; von Savigny et al. (2015)), and Ozone Monitor Profiling Suite Limb Profiler (OMPS-LP; 2011–present; Taha et al. (2021)). Figure 2 shows a 2D histogram of the ratio of measured over interpolated aerosol at 756 nm versus the AE for all data (i.e., all altitudes, locations, and dates available) in the SAGE III/ISS record prior to 2023 (i.e., ratios greater than 1 on the y-axis indicate that AE interpolation creates a negative bias). A running median shown as a solid gray line, along with the plus and minus of the median absolute deviation as dashed gray lines, guides the reader to see how this bias varies with AE. There does appear, at first glance, the possibility of a characteristic behavior between this bias and the AE, but such a relationship is buried in other features. However, it is important to realize that this is completely unfiltered data. Ideally, this analysis would be performed on data that is known to be just aerosol without any potential data quality issues. To this end, Fig. 2 highlights four regions, designated A, B, C, & D, that will be discussed. These regions, as drawn in the figures, roughly correlate to causes of outliers or edges of the distribution of aerosol data such as clouds or noise. The lines delineating them are not precise, and, for some causes, the regions may actually overlap or blur together. They are roughly drawn from experimenting with different filtering criteria and seeing how these criteria affect the overall distribution of data. For the sake of brevity of this paper, the many actual or potential filtering steps are not shown in figures. However, given what will be the broad applicability of the final results of this work, it is important to discuss each region and detail explicitly why some data was excluded.

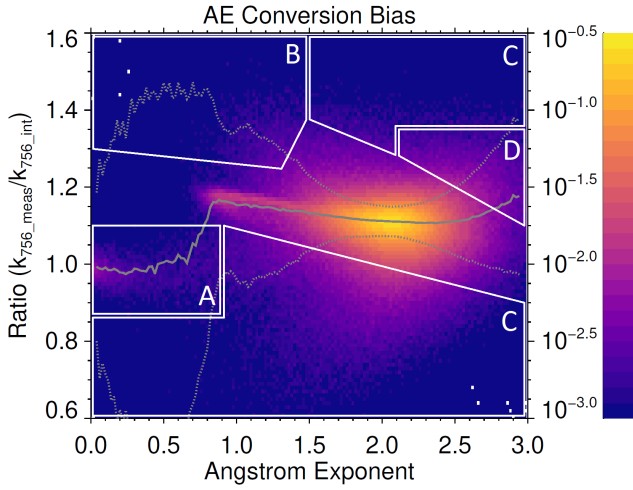

**Figure 2.** Histogram of the bias between the measured and interpolated aerosol at 756 nm as a function of the Ångström exponent. The histogram axis shows the fraction of all events used in this figure, which includes all SAGE III/ISS data between June 2017 and December 2022. The solid gray line illustrates a running median of the histogram and the dashed lines show the median plus and minus the median absolute deviation. The four regions (A, B, C, & D) roughly drawn in white highlight particular regions of interest that correspond to different areas of potential data quality concerns and are described in the text.

Region A is the simplest, as it roughly indicates cloud-aerosol mixtures or measurements that are influenced by the presence of clouds. To avoid potential cloud interference, we used the following simplistic cloud filter. For each profile, first find where $k_{520} \geq 10^{-3}\,\mathrm{km}^{-1}$ and where $k_{520}/k_{1021} \leq 1.75$, then exclude all data at and below the altitude layer located 1 km above this point. These criteria are slight modifications of various previous simplistic cloud filtering techniques applied to SAGE data (e.g., Wang et al., 2002; Davis et al., 2021). The 1 km buffer is meant to avoid any field-of-view or smoothing effects in the SAGE retrieval algorithm. This portion of the data distribution can be more easily seen in Fig. 3, which shows a 2D histogram of $k_{756\_meas}$ versus the AE for all of the same data as Fig. 2 as well as the same highlighted regions as they roughly correlate between the two figures. While more complex techniques exist (e.g., Kovilakam et al., 2023), for SAGE III/ISS data collected up to the end of 2022, this does a reasonable job at removing clouds. However, it is likely that this filter would be too conservative if higher aerosol loading were present in this data.

Region B is both seemingly simple and yet somewhat complicated. Much as with the 384 nm aerosol channel, at sufficiently lower altitudes the 449 nm channel also becomes unusable due to excessive molecular scattering. However, this channel is used for the polynomial fit that is used to first interpolate to 520 nm that is subsequently used to compute AE. Combining this fact with the known deficiencies of occultation geometry in accurately measuring deep into the troposphere mean, for the purpose of this analysis, it may be wise to stick to the stratosphere. As such, a simple filtering of all tropospheric data (with a 1 km buffer above the reported tropopause height) not only eliminates all of the data in Region B, it also helps to ensure the almost complete exclusion of clouds in these SAGE III/ISS data. However, there have been some volcanic and smoke events that have

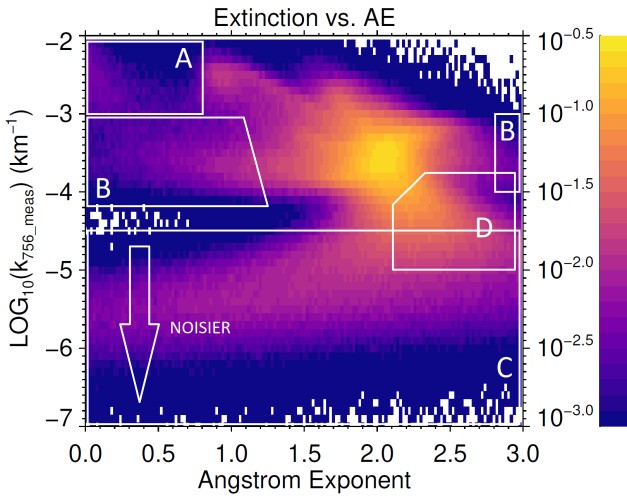

**Figure 3.** Histogram of the measured aerosol at 756 nm as a function of the Ångström exponent. The histogram axis shows the fraction of all events used in this figure, which includes all SAGE III/ISS data between June 2017 and December 2022. The roughly drawn regions in white correspond to those in Fig. 2.

resided very close to the tropopause, so applying this filter does remove some of those data (e.g., the portion of Region B in Fig. 3 that exists at high AE). Still, it does not remove all of that data so excluding tropospheric data does not omit entire significant aerosol loading events. While this filter is effective at removing potentially anomalous data for this study, it does not remove all of it. The "complicated" part of this filter, is that it does not push as close to the core of the distribution as it should, ignoring some data roughly within $1.25 < Ratio < 1.35$ and $1.3 < AE < 2.0$. This will be discussed later on.

Region C takes up a significant portion of Fig. 2, but it represents the noisiest parts of the data. Looking at Fig. 3, we see that all of Region C exists at smaller extinctions, below the core of the overall distribution. The core of this distribution is simply the Junge layer, with all data at generally larger extinctions (excluding Regions A and B) representing specific aerosol loading events and all data at generally lower extinctions representing the data at altitudes near the upper end of or above the Junge layer. In fact, at least at lower AE, there is a clear distinction between the two extinction regimes. Initially we looked at simply filtering based on reported uncertainties. Naturally, using stricter criteria (e.g., exclude all data where uncertainties are >50% as opposed to say >100%) simply incrementally removed more data from the bottom of Fig. 3 and pushed the distribution of data in Region C in Fig. 2 closer to the core of the distribution. Ultimately a filter on uncertainty was abandoned as the filter for Region D also excluded all of Region C, but the data in Region C highlights how a simple filter of uncertainty is often insufficient as a discriminator of aerosol data.

The motivation for investigating Region D came because the running median in Fig. 2 curves up and away at higher values of the AE despite being mostly linear everywhere else. The reason for this is again gleaned when looking at Fig. 3. The bimodality as a function of extinction at any given AE (think vertical cross-section of the figure) is readily apparent. These distributions do start to blend together going from lower AE to higher AE (within the Junge layer) before actually starting to separate again.

Focusing on all data with $AE > 2.5$, a bimodal distribution is still apparent, with larger extinctions mostly representing smaller aerosol particles from the 2018 Ambae eruption (Kloss et al., 2020; Wrana et al., 2023) and smaller extinctions representing smaller aerosol particles above the Junge layer from evaporating sulfate aerosol (Kremser et al., 2016). The tails of these two distributions blend together, with a local minimum at $k_{756\_meas} = 10^{-3.75} \, \mathrm{km}^{-1}$. The portion of the distribution found at lower extinctions is entirely in the upper portions of profiles above the Junge layer (mostly, but not exclusively, at altitudes >25 km) under non-elevated conditions. For this reason, the third and final filter applied here is to exclude all data where $k_{756\_meas} < 10^{-3.75} \, \mathrm{km}^{-1}$, which excludes both Region D and Region C.

The extinction threshold filtering criterion eliminates a significant amount of the data (roughly 20% of data between the tropopause and 25 km), so it is worth further discussing what caveats this creates. Again looking at Fig. 3, this threshold eliminates all non-elevated data above (in altitude) the Junge layer as well as a portion of the core of the distribution (though this portion does not significantly impact the results). Starting from the core of the distribution and moving toward smaller extinctions, two things are apparent: 1) the center of the distribution slowly moves toward larger AE and 2) the spread of the data in AE increases significantly. The slow movement toward larger AE above the Junge layer is expected if aerosol is evaporating and particle size is decreasing (Kremser et al., 2016). This means that the curvature seen in the running median at larger AE in Fig. 2 is potentially real and not an artifact. It also means that the results of this study now apply to fresh aerosol as well as the bulk of the Junge layer, but not the non-elevated data above the Junge layer. Of course, more study may be needed to assess the efficacy of the SAGE III/ISS extinction spectra at these weaker extinctions and higher altitudes to determine why the spectral shape would be different compared to say those smaller particles from a fresh eruption as opposed to a possible data quality artifact. Regarding the increasing spread of the data in AE, particularly toward smaller AE at higher altitudes, it is likely related to the so-called "noise floor" in SAGE observations. This occurs in the region of profiles where any extinction of the Sun is indistinguishable from noise and manifests in the Level 1 data as seeing optical depths asymptote to some small value (both positive and negative). Because of the smoothness of this feature with wavelength, the SAGE retrieval algorithm tends to interpret the noise floor amplitude as aerosol (with smaller AE). This means that the magnitude of aerosol extinctions, at higher altitudes, can asymptote to some small value (typically averaging data will reduce or even remove this effect), which will negatively impact the ability to compute AE or interpolate between channels. While further study may be needed, the overall spread of the data in AE at lower extinctions is likely a combination of the noise floor effect and increasing uncertainties that may smear out this effect.

## 4  Results

Applying the three filtering criteria described above (i.e., clouds, troposphere, and lower extinction) to Fig. 2 yields Fig. 4. With most potential data quality problems (and highly-aged aerosol) filtered out, the running median of the AE interpolation bias exhibits a clear linear behavior. In fact, a straight line fit in red over all data with $0.9 \leq AE \leq 2.2$ still follows the running median at both lower and higher AE. The slope and intercept of this line are also shown. These values can thus provide a simple correction to using the AE formula for conversion of aerosol extinction data from one set of wavelengths to another using the

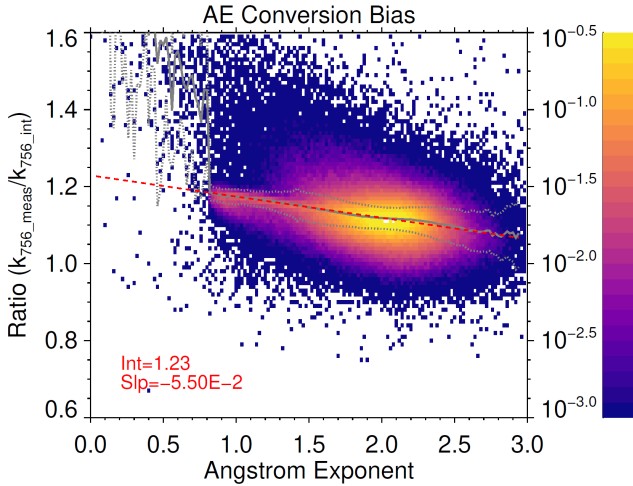

**Figure 4.** The same as Fig. 2 after applying the filtering described in the text. The histogram axis still shows the fraction of all events used in this figure, but the total number of data points is smaller than in Fig. 2. A straight line fit of the data between $0.9 \leq AE \leq 2.2$ is shown along with the slope and intercept of the fit.

following simple conversion:

$$k_{corr} = k_{AE}(m \times AE + b), \tag{2}$$

where $m$ and $b$ are the slope and intercept, respectively, from Fig. 4, $AE$ is the Ångström exponent computed using the wavelength pair of 520 and 1021 nm, $k_{AE}$ is the extinction computed using the AE formula (i.e., Eqn. 1), and $k_{corr}$ is the corrected value of the extinction after accounting for the AE interpolation bias. However, despite the behavior being generally

well-characterized by a straight line, there is still a branch of the overall distribution that does not follow this. This is the data that was discussed earlier as "complicated" regarding Region B in Fig. 2 (roughly $1.25 < Ratio < 1.35$ and $1.3 < AE < 2.0$). It turns out this branch is another data quality issue that is currently intrinsic to v5.3 of SAGE III/ISS data. While filtering out tropospheric data removes most of this branch, it does not remove all of it and the reason is because it is a positive bias in the 756 nm channel as a result of spectral stray light from the nearby oxygen A-band. This was actually first noticed in

Wang et al. (2020) and again recently in Boone et al. (2023), but has not yet been fully investigated or accounted for in the SAGE III/ISS algorithm. Effectively, the point spread function of the 756 nm channel is sufficiently wide to pick up some attenuation present in the A-band, but it is not removed in processing. An ad-hoc filtering of the aerosol could be applied by excluding all data with sufficiently high neutral density (since A-band absorption scales with neutral density), but this begins to remove too much legitimate aerosol data. Fortunately, doing so does not appreciably change the slope and intercept shown

in Fig. 4. To further highlight that this actually a bias in the data and not some secondary branch of aerosol that could be from some other composition, Fig. 5 shows the same data as Fig. 2 except for the 869 nm channel. Here, where there is no known reason for spectral stray light to create a positive bias, this branch in the bias distribution does not exist. It is worth noting that this positive bias in the 756 nm channel will be mitigated in a future version of the SAGE III/ISS data.

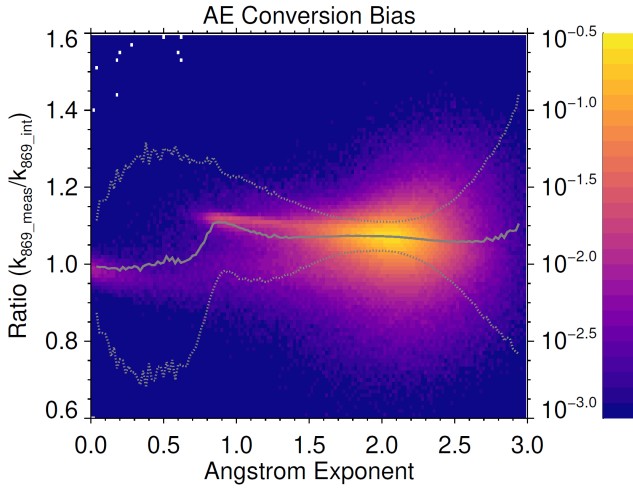

**Figure 5.** The same as Fig. 2 except for data at 869 nm.

It is useful to look at these particular results in the context of prior work. Two previous studies computed aerosol extinction coefficients using PSD parameters combined with Mie theory to briefly look at the AE interpolation bias at 750 nm when using 525 and 1020 nm to compute the AE: Rieger et al. (2015) and Malinina et al. (2019). Rieger et al. (2015) calculated the theoretical interpolation bias for a range of mode radii and mode widths, but only showed biases smaller than 15% for large particles (see their Fig. S2), whereas Fig. 4 shows biases as large as 20% for some aerosol after the Hunga-Tonga eruption and hints at potentially larger biases for smaller AE. Malinina et al. (2019) used PSD parameters derived from satellite-based limb scatter measurements combined with Mie theory to compute extinctions at 525, 750, and 1020 nm and compared them with AE-interpolated extinctions at 750 nm, but showed an altitude-consistent bias of around 8% over the period of 2002–2012 (see their Fig. 6). However, this time period did not see significantly large or sustained aerosol loading events and their derived AE showed a large, altitude-dependent bias with SAGE II measurements (see their Fig. 7). Even so, considering a value of AE in between what that study called "background" and "unperturbed" of 2.5 (see their Table 2) and using the empirical correction described by Eqn. 2 for 756 nm yields a bias correction of about 9.3%. While a direct comparison between this work and previous studies is not possible, it seems that theoretical Mie-based corrections for the AE interpolation bias are smaller than, though still fairly consistent with, those derived here from SAGE III/ISS measurements. It is worth noting that the AE-interpolated extinction bias will inherently have some spread for a given AE due to variability in the PSD. However, the observed spread in the distribution shown in Fig. 4 is well described by the reported uncertainties in the data (analysis not shown here). Thus, the spread in interpolated extinction coefficient at any given AE due to variability of the PSD is likely significantly smaller than both what is shown in this work and what has been predicted by previous studies for at least the aerosol loading conditions and wavelengths shown here. While the results between theory and observation can be fairly similar, the differences are still noticeable. Attempting to reconcile any discrepancies would be an important area for future study, though it is not unreasonable to speculate that some of the assumptions previously used for theoretical calculations

**Table 1.** Slopes and intercepts of the straight line fits to the AE conversion bias at different SAGE III/ISS wavelengths when AE is computed using 520 and 1021 nm.

| Wavelength | Slope [Ratio/AE] | Intercept [Ratio] |
|---|---|---|
| 449 nm | $+4.44 \times 10^{-2}$ | 0.80 |
| 602 nm | $-3.69 \times 10^{-2}$ | 1.15 |
| 676 nm | $-5.37 \times 10^{-2}$ | 1.22 |
| 756 nm | $-5.50 \times 10^{-2}$ | 1.23 |
| 869 nm | $-4.26 \times 10^{-2}$ | 1.16 |

may need refinement. For example, the oft-used assumption of a unimodal log-normal distribution for modeling the aerosol particle size has recently shown to be less compatible with observations than a bimodal distribution (Boone et al., 2023). In that study, observed extinction spectra combined across both visual and infrared wavelengths were fit with both modeled unimodal and bimodal distributions and compared. Not only did their example model fits to transmission profiles (their Figs. 2 and 4) show that bimodal distributions fit the observations better (with different curvatures in the modeled spectra in each case), they also indirectly suggest that using a unimodal distribution would result in a smaller AE interpolation bias than shown by observations. While not a straightforward comparison, the unimodal distribution model fits (panel A in both figures) reveal a positive bias in extinctions at 520 nm (i.e., negative bias in transmission) compared to observations, which would cause the AE interpolation bias at 756 nm to become smaller. One way to visualize this would be to look at Fig. 1 in this work and observe how increasing extinctions at 520 nm relative to 756 nm would cause the dashed lines (i.e., AE interpolation) to get closer to the solid lines (i.e., actual extinctions). Naturally, using data and measurements to improve our understanding of the assumptions that are often placed on aerosol parameters is an extremely important topic and perhaps these results may be useful in such an endeavor.

Ultimately, the purpose of this study was to highlight the fact that using the AE to interpolate between aerosol channels introduces a bias. Fortunately, the multispectral aerosol extinction data from SAGE III/ISS enables the quantification of this bias. The behavior of the bias is surprisingly robust across altitudes, latitudes, extinction levels, and compositions. It is straightforward to repeat this analysis and compute the slopes and intercepts for correction of the AE interpolation bias for other SAGE III/ISS wavelengths. Table 1 provides these values at five SAGE III/ISS wavelengths when the AE is computed using the 520 and 1021 nm channels. These values use the same filtering criteria as already discussed, except that the extinction threshold filter value is slightly different for each wavelength ($10^{-3.30}$ km$^{-1}$ for 449 nm, $10^{-3.55}$ km$^{-1}$ for 602 nm, $10^{-3.65}$ km$^{-1}$ for 676 nm, $10^{-3.75}$ km$^{-1}$ for 756 nm, and $10^{-4.00}$ km$^{-1}$ for 869 nm). In theory, this work could be repeated using other possible wavelength pairs to compute the AE such as the combination of 756 and 1544 nm. Given the various filtering criteria used to create Fig. 4, it is important to discuss the applicability of Eqn. 2. The obvious exclusions are that Eqn. 2 does not apply to clouds or tropospheric data. The minimum extinction threshold filter does create a caveat, namely that the very

small particles in the upper stratosphere that likely result from evaporating aerosol are not completely characterized by Eqn. 2. However, since that data exists at ratios greater than 1 (Fig. 4) and the fit of Eqn. 2 does not cross a value of 1 until the AE approaches 4, the correction is still in the right direction so there is no harm in applying it to aerosol data that falls within this category. As such, Eqn. 2 can be considered applicable to all cloud-free stratospheric data with $AE \lesssim 4$ (varies slightly with the wavelength chosen) with the understanding that some bias independent of this correction will still be present in the regime of evaporating aerosol in the upper stratosphere. Additionally, provided that the measurements from SAGE III/ISS are a reasonable representation of aerosol, these results should be considered applicable to data from any instrument and measurement technique.

The slopes and intercepts defining the empirical relationship of the AE interpolation bias behavior can be used when converting aerosol from other instruments when measurements of the desired wavelength are not available. This correction can improve efforts of merging aerosol data from multiple instruments such as the Global Space-based Stratospheric Aerosol Climatology (GloSSAC; Kovilakam et al. (2020)) and the Climate Data Record of Stratospheric Aerosols (CREST; Sofieva et al. (2024)). These results may also be helpful in the context of comparisons of current and future instruments that measure aerosols at different wavelengths. While SAGE III/ISS and OMPS-LP are still operating, there are plans for future satellite-based occultation, limb scatter, and lidar instruments that will observe aerosol at a number of different wavelengths that are the same or similar to the wavelengths reported by SAGE III/ISS. Proper conversion of aerosol data between wavelengths will be necessary to compare and validate these instruments.

## 5 Conclusions

An empirical correction for the bias introduced when interpolating aerosol between two wavelengths using the Ångström exponent has been demonstrated using multispectral measurements from SAGE III/ISS. These biases are the result of the fact that the aerosol extinction spectrum does not strictly follow the Ångström exponent formula that is commonly used for this purpose. The correction is generally applicable to all non-cloud stratospheric aerosol, regardless of composition, with the minor caveat that it is possible it does not apply to highly-aged evaporating aerosol above the Junge layer, though more study is needed to see if this exclusion is a data quality artifact. The results of this work suggest that past modeling studies of this effect may underestimate the magnitude of the correction, potentially highlighting that some of the assumptions used for those studies may need refinement. The correction was evaluated at 5 SAGE III/ISS wavelengths (449, 602, 676, 756, and 869 nm) when the AE is computed using 520 and 1021 nm and can be used for any future attempts to convert aerosol data from one wavelength to another using the AE interpolation method. This correction will hopefully be useful for intercomparisons of past, current, and future aerosol instruments that utilize different measurement techniques at different wavelengths as well as for improving the merging of multi-instrument aerosol climatologies.

*Data availability.* SAGE III/ISS v5.3 data (DOI:10.5067/ISS/SAGEIII/SOLAR_BINARY_L2-V5.3) are publicly available from the NASA
Atmospheric Science Data Center at https://eosweb.larc.nasa.gov/ (last access September 2023).

*Author contributions.* VS and AR motivated the study through their work on the CREST data set and performed some initial analyses. RD
incepted the study and performed much of the analysis. LT helped refine and explain the filtering criteria. All authors contributed to the
manuscript.

*Competing interests.* The authors declare that they have no conflict of interest.

*Acknowledgements.* SAGE III/ISS is a NASA Langley managed mission funded by the NASA Science Mission Directorate within the Earth
Systematic Mission Program. Enabling partners are the NASA Human Exploration and Operations Mission Directorate, the International
Space Station Program, and the European Space Agency. The ongoing development, production, assessment, and analysis of SAGE data sets
at NASA Langley Research Center is supported by NASA's Earth Science Division. VS and AR thank the ESA project CREST. VS thanks
the Academy of Finland (Centre of Excellence of Inverse Modelling and Imaging; decision 353082). AR was funded in part by the University
and the State of Bremen.

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
