# Peer review of "An Empirical Characterization of the Aerosol Ångström Exponent Interpolation Bias using SAGE III/ISS Data"

_Atmospheric Measurement Techniques, 2023_

## Author Comment (AC1)

The authors would like to thank the referee for taking the time to review this paper and for the many helpful comments that will be used to improve it. The referee's comments/concerns are listed below in red text, while the authors' responses to each comment are written below in black text.

The manuscript deals with an important problem and provides a lot of useful analysis on the SAGE III data, but in my opinion motivation for the project and how these corrections are intended to be used needs to be expanded upon. I believe the main motivation for this work is so that SAGE II can be brought to ~750 nm for CREST/GLoSSAC, but this is never explicitly stated. I understand the authors wish to present a general technique, but many choices made by the authors are directly motivated by this specific application (e.g. the AE wavelengths, correction factors only given at the SAGE III wavelengths) chosen, and unless the reader knows this beforehand it can be hard to follow. As a reader I cannot think of another application beyond this at least for the current instruments we have, if the authors have other applications in mind they need to be explicitly stated. To be clear, I believe the conversion of SAGE II is a very important application, but the authors should provide concrete (rather than vague) examples of how this work is going to be used early in the manuscript to help the reader understand the authors choices.
p.9 l 188 "Repeating this analysis for multiple channels provides slopes and intercepts of this behavior that can be used when converting aerosol from other instruments when measurements of the desired wavelength are not available. ..."
- Here and the rest of the paragraph is when the motivation for this work is given, some of this information similar needs to be in the introduction/abstract.
p. 9 l 196 "In theory, this work could be repeated using other possible wavelength pairs to compute the AE such as the combination of 756 and 1544 nm."
- I understand the author's desire to present a general technique, but do the authors have another application in mind besides converting SAGE II using the 520/1021 AE?

We tried to add mention of the motivation behind this work (i.e., to aid in the conversion of aerosol data between wavelengths for the purpose of instrument intercomparisons and the improvement of multi-instrument climatologies) in the abstract, introduction, results, and conclusions.

p. 2 l 40 "The only assumption used in the retrieval is that the "aerosol spectrum" (i.e., extinction as a function of wavelength) should be slowly varying in almost all stratospheric conditions"
- The authors probably mean the only assumption on aerosol microphysical parameters or something similar since there are many assumptions made in the SAGE III retrieval.

This has been changed to "The only assumption regarding aerosol used in the retrieval"

p. 3 l. 55 "For this study AE is evaluated using the 520 and 1021 nm channels as this is one of the most common pairs for evaluating the AE using SAGE data"
- Going back to my general comments, on the first read through I did not pick up that here SAGE is meant more generally, i.e. for SAGE II. And as far as I can tell this is the only hint so far as to what this correction is actually going to be used for at this point.

We try to be explicit, which is why "SAGE III/ISS" is written out so much (or sometimes "SAGE II") instead of just shortening it to "SAGE" such that "SAGE" refers to any SAGE instrument.

p.3 l.63 "since at lower altitudes the signal in this channel drops significantly from molecular scattering..."
- The same dip is also visible in the highest altitude in the picture (23 km) during the Hunga-Tonga eruption, so it can be caused by molecular scattering or large aerosol optical depth. For the Hunga-Tonga time period in particular, is it possible that the signal due to direct forward scattering is no longer insignificant as assumed by SAGE III?
We have added additional wording to better clarify this. The "dip" we refer to (specifically using the word dip in quotations and described at the end of Section 2) is seen as the difference between exes and squares in Fig. 1 in the 520, 602, and 676 nm channels. The large bias in the 384 nm channel is again shown as the large difference between exes and squares. In this case, a large drop/bias is not seen at higher altitudes, only lower altitudes as a result of a very weak signal because molecular scattering is so high. What the reviewer is referring to at higher altitudes is a change in the shape of the aerosol spectrum as a byproduct of changes in particle size, but notice how the exes and squares are much better aligned at 384 nm at the three higher altitudes when compared to the three lower altitudes.

p.4 l 85 "The 1 km buffer is meant to avoid any field-of-view or smoothing effects."
- Has the SAGE III data been post-processed with one of the standard altitude smoothing filters?
The SAGE III/ISS data used here has not been post-processed with any smoothing. The smoothing to which we refer has to do with the Level 1 retrieval algorithm that provides some smoothing of packet-level data during processing of transmission data prior to the vertical binning process that produces transmission data on a standard grid at 0.5 km intervals. We have noticed that SAGE (II & III) can sometimes have some blurring of the final data product in the presence of strong vertical gradients such as from clouds.

p.6 l 135 "extinction of the sun"
- Does this mean the solar only spectra that is used for normalization?
The noise floor applies to data that is not just exoatmospheric (i.e., where there is no net extinction/attenuation). Basically, the instrument measures uncalibrated intensity represented as counts in the detector. These count values will fluctuate even when staring at some uniform source. If the source dims by some amount smaller than the amplitude of the fluctuations, it can be difficult to differentiate this dimming from the noise. Effectively, the instrument can only reliably measure optical depths in the attenuated atmosphere that are so small.

p.8 l 179 "it seems that theoretical Mie-based corrections for the AE interpolation bias are smaller than, though still fairly consistent with, those derived here from SAGE III/ISS measurements."
- This should be expanded upon. Mie scattering is the fundamental physics behind these errors, are you saying that these previous corrections did not include some factor (large enough particles, correct composition) or are you saying that there is a potential bias in the SAGE III extinction spectrum? I'm sure as the authors know, there are a few recent studies on deriving

aerosol microphysical parameters from SAGE III extinction spectra, this could have implications for that.

It is honestly impossible to say for certain whether the discrepancies between previous Mie-based studies and this SAGE measurement-based study come from a "fault" in the former or the latter or perhaps a bit of both. While there is some inherent uncertainty in the measurements, they are based on observations and retrievals that make no real inherent assumptions about the properties of aerosol. The theory-based studies, however, must make some assumptions about aerosol microphysical parameters. We do not currently have any reason to believe in any biases in the SAGE data that would affect the conclusions of this study at this time. It would require a far greater effort to attempt to reconcile these discrepancies than is within the scope of this study, but it is true that using data and measurements to improve our understanding of the assumptions that are often placed on aerosol parameters is an extremely important topic. We have added some text to this paragraph and would welcome the reviewer's feedback.

Entire manuscript-> In many cases the units for extinction are missing
The units for extinction have been added throughout the text.

---

## Author Comment (AC2)

The authors would like to thank the referee for taking the time to review this paper and for the many helpful comments that will be used to improve it. The referee's comments/concerns are listed below in red text, while the authors' responses to each comment are written below in black text.

A lot of information and motivations seems to be omitted in the text and it is implied that reader knows that information or assumes from the context. As a result, it makes the paper very hard to comprehend, in particular on the first read. A few examples:
**a)** In discussion of Fig. 3 and Fig. 4, neither of which are not temporally resolved, authors mention Ambae (p.6, l. 117) and Tonga (p. 8, l. 172) eruptions, which is implied that the reader knows of the magnitude of the aerosol extinction and AE and can easily identify them at the extinction/AE plot. While Tonga eruption was extraordinary, I doubt that even stratospheric aerosol specialists will be able to point that eruption on those plots. On the other hand, Ambae eruption was large, but there were other events which were of comparable magnitude during SAGE III operation time (e.g., 2019 Raikoke eruption), so again, on the 2D extinction/AE histograms these eruptions are extremely hard to identify. As a solution I suggest to either specifically circle those regions on the plots or provide the extinction and AE value ranges for these eruptions.

The specific parts of the distribution attributable to each event is not really all that pertinent to the main message of this paper. In particular, Fig. 2 does not appear all that different essentially if you time resolve it as all of the data tends to follow the distribution shown (depending upon what the stratospheric aerosol load is at that particular time). However, we have taken Figs. 2 and 3 and recreated them for different 4 month periods surrounding events of interest and added it to a supplement.

**b)** Another example, relates to the knowledge of occultation technique. In the first sentence of Sec. 2 it is mentioned that occultation technique "intrinsically provides vertical profiles of extinction data", which is absolutely true; however, a little more information on the technique should be given. If authors are concerned about the length, then at least some references should be provided.

We have added some minor wording additions and references.

**c)** I couldn't find explicit information on the the extent of altitudes used for Figs. 2-5 (and as a result for Eq. (2)). Based on the text of the whole paper, I assume it is some range between surface and 17 km as the lowest border and 45 km as the top since clouds and small particles above Junge layer are discussed. Please, provide this information since it is extremely important for interpretation of the results.

The paper mentions that Fig.2 uses "all data in the SAGE III/ISS record prior to 2023", which was meant to imply data at all altitudes, locations, and dates available. We have added a statement to be more explicit.

Maybe this comment will arise from the previous, but I did not quite understand the purpose of the discussion of the regions A-C in Sec. 2. While it is nice to see why people did what they did in the conference talks, here this part makes the paper unnecessarily long and harder to read. As far as I could tell the outcome of almost two page of the description of regions A-C is: the

clouds, tropospheric aerosols and data with large uncertainties were filtered out (actually through region D, not through region C). Clouds are a known problem in the stratospheric aerosol retrieval community (cited in the paper Rieger et al. (2019), Kovilakam et al. (2023) or any other paper on stratospheric aerosol retrieval). Similarly, justification for not including tropospheric data could be summarized more briefly. While I might be wrong in my interpretation of the importance of this part, in the current form in my opinion these two pages could be summarized in a couple of sentences without lengthy descriptions. If authors agree with this comment, but would like to keep the descriptions, they could move them into supplement, otherwise context needs to be provided.

Respectfully, the importance of "showing all your work" should be in every paper (given the space when not restricted by the journal) as opposed to conference talks where time is limited and brevity is key. It is true that problems with clouds and/or tropospheric data are sufficiently common that we could have simply stated we excluded these data and been done with it. However, because the final result is so widely applicable, we thought it best to illustrate all of the data and highlight why some data were excluded. Now Regions A and B (and most of C) do not take up much room in the paper despite the theoretical possibility of explaining them away in a single sentence. The discussion of the upper (i.e., larger extinction) portion of Region C as well as Region D is more nuanced and requires detailed explanation and justification as it then creates a potential caveat on the use/interpretation of the results.

While the derived Eq. (2) is highly important, and I anticipate it to be used by many, I think the discussion of its transferability and limitations needs to be included. Few things come in mind: **a)** Related to the last bullet in the first major comment is the formula independent of altitude, or is there a specific altitude range where it can be used? Or only low threshold on the extinction coefficient mentioned at p.9 l.195-196 matters? Is there an upper threshold?

In general, the results of Eq. 2 (and Table 1) are widely applicable to all aerosol data in the stratosphere. While the discussion of Region D does leave a bit of a caveat, namely that the very small particles in the upper stratosphere that (likely) result from evaporating aerosol are not completely characterized by Eq. 2, the correction is still in the correct direction so there is no reason not to apply it. It will just be important to understand that some bias from using the "corrected" form of AE interpolation will still be present in this regime. We have added these clarifications to the text.

**b)** This formula was derived for 756 nm from 520/1021 AE. Is this formula transferable to the other instruments (e.g., SAGE II, SCIAMACHY) which operated at the same wavelengths but under different stratospheric conditions? Can it be used for occultation or it is applicable to limb scattering instruments too?

Given how the data appears to conform to this relationship across the wide range of events in the SAGE III/ISS record, we believe this formula should be applicable regardless of the stratospheric condition. It can certainly be used for occultation, though there is a pseudo caveat for its use with limb scatter. Given the reliability of the occultation technique, it is likely that the occultation data is a reasonable representation of the behavior of this correction as it essentially applies to the shape of the aerosol spectrum. However, it is also likely that the assumptions that go into limb scatter retrievals can/will produce biased aerosol extinction data under certain loading conditions. Depending upon the nature of this bias, applying this correction may yield results that agree "better or worse" when compared with occultation data. However, it is

important to remember that the retrieval biases and this empirical correction are independent. Thus, we would recommend using this correction to limb scatter data when applicable and addressing any retrieval biases separately.

**c)** While the formula was derived for the SAGE III data, would it be the same if calculated with Mie theory using a plausible range of particle sizes (e.g., using the cited in the text Wrana et al. (2021) product)? There is a discussion about the other authors getting much lower uncertainties with Mie theory (p.8 l.168 - p.9 l.185) using SAGE II, but I was wondering why is that so? Is it because the assumption on the unimodal log-normal distribution made by Rieger et al. (2015) and Malinina et al. (2019) is incorrect (possible, and that would be an important statement)? Or is it some sort of SAGE III instrument-specific feature (which is also possible)? Maybe it does not make sense to compare directly with the other authors' results, but it is quite feasible to do this simulation for SAGE III.
While I understand that covering all those point in details might be enough for another paper, at least acknowledging those issues would be crucial for the formula's future application.

The formula is effectively related to the shape of the aerosol spectrum, which is ultimately dependent upon the aerosol properties. Computing it from Mie theory will thus be dependent upon the assumption of those properties, which can be highly varied. One such assumption, as the reviewer suggests, regarding the unimodal log-normal distribution has already recently been shown to be likely in need of refinement (see Boone et al., 2023). In fact, a brief look at the figures of modeled transmission from that work suggest that the unimodal distribution assumption may lead to a reduced AE interpolation bias. As we responded to Reviewer 1, we do not currently have any reason to believe in any biases in the SAGE data that would affect the conclusions of this study at this time. As the reviewer points out, reconciling the discrepancies between theory and observation is definitely a work unto itself and better suited for another paper. Still, we have added some text to the paragraph regarding the comparison between this work and previous studies and welcome the reviewer's feedback.

Throughout the text the term "aerosol(s)" is used as a substitute term for "stratospheric aerosols" and while for certain contexts it is fine (p.4 l.81), in some it leads to misleading statements (e.g., p2. l. 30-32). Please, correct throughout the text.

We have added a statement at the beginning of Section 2, after changing "aerosol" to "stratospheric aerosol" that we will thereafter simply refer to "stratospheric aerosol" as "aerosol" for brevity.

P.1 l.23. "aerosol spectrum" is a jargonism, it is defined at p.2 l.41. Either define here, or change to "extinction as a function of wavelength" or similar.

We have moved the definition to the first instance of this terminology.

P.2 l.30. Please, add "space-borne" before instrument and "stratospheric" before aerosol. Otherwise, you summarize even over such measurements like AERONET (Holben et al., 1998) for the total column.

We have made the previously mentioned change regarding the term "stratospheric" and also included the term "remote sensing" before "instruments". The initial statement of this paragraph is with regard to vertical profiles of aerosol extinction (not total column) being intrinsically measured by the occultation technique.

P.2 l.41. It is not obvious what you mean under "should be slowly varying in almost all stratospheric conditions".

Effectively it means that, unlike trace gas species, there are no noticeable absorption features and the scattering does not change rapidly with wavelength either. At SAGE measurement wavelengths, this is a reasonable assumption.

P.2 l.53. Malinina et al. (2019) cited in this paper a few times showed that the AE on one wavelength pair is not an indicator of particle size. Please rephrase this statement.

That is true, but the AE (or "color ratio") is often referred to in that context. There is a general correlation between AE and particle size as it pertains to volcanic eruptions and SAGE data that is informally used in the literature or in conversation. This statement is not meant to be a definitive one (i.e., one value of AE does not equate to one precise PSD), rather mentioning how AE is often, even if informally, referred to. We have modified the text slightly to: "…often acting as an ad-hoc (albeit imprecise) indicator of effective particle size …"

Fig. 1, please add years of the eruptions and wildfires.

We have added the years of the events as well as various references to them in the text.

P.3 l.61-62 vs Fig. 1. From what I can tell from the Fig. 1, the "dip" is not always an artifact with bias being present just in half of the shown spectra, which indicates that there is some information in there. Can you please elaborate?

The "dip" only refers to a bias in the 520, 602, and 676 nm channels. This was shown much more clearly and explicitly in Wang et al. (2020). It is still a matter of investigation for the SAGE team, though we have reason to believe it is related to ozone spectroscopy. Its actual behavior can vary from event to event, as well as with altitude and aerosol loading conditions. Sometimes it is not as noticeable, particularly in averaged data. It is only slightly noticeable in some of the "spectra" shown in Fig. 1, but rest assured it is there. Note this is different from the bias seen in the 384 nm channel at low altitudes that is the result of a loss of signal in the instrument as a result of very large molecular scattering. We have also made sure to add clarification to what we are referring to in the text.

P.3 l.69. I would list the instruments with the retrieved products at 750 nm.

Added

P.4 l.86. Technically, in Fig. 3 you show log($k_{756\_meas}$).

Added

Color bars labelling in Figs. 2-5 is needed (I assume it is probability density).

The figure caption states that the "histogram axis shows the fraction of all events used in this figure" so the color bar is a fraction of total events.

P. 8 l. 172: "Tonga aerosol" is a jargonism. Also, please, use consistent naming for the volcano (in Fig. 1 it is "Hunga-Tonga"). Maybe, "aerosols after the 2021 Hunga-Tonga eruption"?

Wording has been changed to be consistent and clearer.

Section 5. Please, update based on the major comments on limitations.
Some wording changes and additional text were added to the conclusion.